# DISTILLM-2: A Contrastive Approach Boosts the Distillation of LLMs

**Jongwoo Ko** [* 1] **Tianyi Chen** [2] **Sungnyun Kim** [1] **Tianyu Ding** [2] **Luming Liang** [2] **Ilya Zharkov** [2] **Se-Young Yun** [1]

[1]KAIST AI    [2]Microsoft    [*]Work done as a research intern at Microsoft

https://github.com/jongwooko/distillm-2

## Abstract

Despite the success of distillation in large language models (LLMs), most prior work applies identical loss functions to both teacher- and student-generated data. These strategies overlook the synergy between loss formulations and data types, leading to a suboptimal performance boost in student models. To address this, we propose **DISTILLM-2**, a contrastive approach that simultaneously increases the likelihood of teacher responses and decreases that of student responses by harnessing this synergy. Our extensive experiments show that DISTILLM-2 not only builds high-performing student models across a wide range of tasks, including instruction-following and code generation, but also supports diverse applications, such as preference alignment and vision-language extensions. These findings highlight the potential of a contrastive approach to enhance the efficacy of LLM distillation by effectively aligning teacher and student models across varied data types.

## 1. Introduction

Large language models (LLMs) have continuously improved their text generation abilities by increasing the number of parameters and the amount of high-quality training data. However, LLMs typically require extensive computational resources during inference, which makes them difficult to be deployed practically. Therefore, compressing them by reducing the number of parameters while maintaining their performance becomes important for using these powerful models effectively.

[1]KAIST AI, Seoul, Republic of Korea [2]Microsoft, Redmond, Washington, USA. Correspondence to: Jongwoo Ko <jongwoo.ko@kaist.ac.kr>, Tianyi Chen <tianyi.chen@microsoft.com>.

*Proceedings of the $42^{nd}$ International Conference on Machine Learning*, Vancouver, Canada. PMLR 267, 2025. Copyright 2025 by the author(s).

As the demand for reducing computational overhead grows, knowledge distillation (KD; Hinton et al. 2015) has emerged as a promising technique for compressing LLMs into more lightweight student models. By transferring knowledge from a high-capacity teacher model to a smaller student model, KD can significantly improve the performance of small language models (sLMs) as demonstrated by Llama 3.2 (Meta, 2024) and Gemma-2 (DeepMind et al., 2024).

Over the years, research on LLM distillation has largely focused on either by designing new loss or by curating training data. From a loss perspective, several studies suggest that Kullback-Leibler (KL) divergence, a common loss for KD, may fail to capture the teacher model's complex generative behavior (Wen et al., 2023; Gu et al., 2024). Consequently, alternative loss functions, such as skew KL (SKL; Ko et al. 2024), have been proposed to better guide the student. On the other hand, from data perspective, previous works emphasize how the training data is curated to enlarge the effectiveness of KD. For instance, relying solely on offline data (*e.g.,* teacher-generated outputs; TGOs) can be problematic where student's outputs at inference time deviate significantly from fixed training samples (Agarwal et al., 2024). To address this mismatch, some works incorporate student-generated outputs (SGOs) directly into training (Lin et al., 2020; Xu et al., 2024b). However, these works often overlook the synergy between loss formulations and data types, which might have limited the extent of performance improvement of student models.

Recently, contrastive approaches such as direct preference optimization (DPO; Rafailov et al. 2023), have gained popularity for their efficacy and efficiency in preference alignment (Tajwar et al., 2024) or reasoning (Pang et al., 2024), by explicitly employing different learning strategies to handle two distinct responses. Despite their success, few works have focused on extending their schema to KD for LLMs. While Li et al. (2024b) attempted to simply apply DPO by replacing the reference model to teacher model (see Equation 4), we observed that their method is prone to reward hacking, which may limit its broader applicability (see Figure 1). This motivates us to design a scalable contrastive approach to boost LLM distillation.

**Contributions.** In this paper, we introduce DISTILLM-2, which features a novel contrastive approach for KD of LLMs. Our DISTILLM-2 builds a contrastive framework upon DistiLLM (Ko et al., 2024), which has shown significant improvements by using SKL-based loss and balanced SGOs. Our detailed contributions include:

- **Contrastive approach with asymmetric loss dynamics:** We analyze the behavior of forward and reverse KL (and SKL) during the training process on responses from the student and teacher models, respectively. This analysis motivated the development of a contrastive approach for LLM distillation (CALD; §3.1), which applies distinct loss functions to different types of training samples. By doing so, CALD effectively incorporates the synergy between loss formulations and data perspectives.

- **Development of the contrastive approach:** Additionally, we introduce optimized dataset curation strategies (§3.2) and curriculum-based adaptive loss mechanisms (§3.3). These enhancements to CALD, which are collectively coined to as DISTILLM-2, provide solid guidelines for our contrastive approach for practitioners.

- **Advanced performance and versatility:** DISTILLM-2 achieves state-of-the-art performance for sLMs across various text-generation tasks, including instruction-following, mathematical reasoning, and code generation (§4). Furthermore, we demonstrate the diverse applications of our proposed KD approach (§6), such as preference alignment with better reference models and its expansion to vision-language models.

## 2. Backgrounds

### 2.1. Related Work

KD (Hinton et al., 2015) effectively compresses neural networks, enabling smaller student models to match the performance of larger teachers. This technique recently has been adapted to address the scalability challenges of LLMs, enhancing their viability in compute-intensive environments. ImitKD (Lin et al., 2020) demonstrated the use of SGO as training data for distillation. Building on this, Agarwal et al. (2024) introduced an on-policy approach with objectives like reverse KL or Jensen-Shannon divergence (JSD). Wen et al. (2023) explored various f-divergences, including total variation distance and JSD, in auto-regressive LMs, while Gu et al. (2024) proposed a policy gradient method to mitigate high variance in RL-based techniques. Recently, Xu et al. (2024b) combined static datasets with on-policy methods using speculative decoding for training data generation. Among these, DistiLLM (Ko et al., 2024) achieved state-of-the-art performance and greater efficiency by introducing SKL and an adaptive off-policy approach. A more discussion of related works is available in the Appendix A.

### 2.2. Preliminary: KD in LLMs and DistiLLM

**Loss function of KD in LLMs.** Given a prompt and response pair, denoted as $(\boldsymbol{x}, \boldsymbol{y})$, KD minimizes divergence between the distributions of a teacher $p(\boldsymbol{y}|\boldsymbol{x})$ and a student $q_\theta(\boldsymbol{y}|\boldsymbol{x})$ parameterized by $\theta$. Conventionally, KL, denoted as $D_{\mathrm{KL}}$, is the most widely used loss in KD due to its simplicity and tractability. The sequence-level distillation using KL is accurately decomposed into a sum of token-wise distillation (Ko et al., 2024):

$$D_{\mathrm{KL}}(\boldsymbol{x}, \boldsymbol{y}; p\|q_\theta) = \sum_{t=1}^{T} p(y_t|\boldsymbol{y}_{<t}, \boldsymbol{x}) \log \frac{p(y_t|\boldsymbol{y}_{<t}, \boldsymbol{x})}{q_\theta(y_t|\boldsymbol{y}_{<t}, \boldsymbol{x})}. \tag{1}$$

We can also define reverse KL as $D_{\mathrm{RKL}}(\boldsymbol{x}, \boldsymbol{y}; p\|q_\theta) = D_{\mathrm{KL}}(\boldsymbol{x}, \boldsymbol{y}; q_\theta\|p)$. Despite its tractability, such KL has limitations of either mode-averaging or mode-collapsing for forward and reverse version, respectively. To address this issue, Ko et al. (2024) proposed skew KL (SKL) and skew RKL (SRKL), defined as follows:

$$D_{\mathrm{SKL}}^{(\alpha)}(\boldsymbol{x}, \boldsymbol{y}; p\|q_\theta) = D_{\mathrm{KL}}(\boldsymbol{x}, \boldsymbol{y}; p\|\alpha p + (1 - \alpha)q_\theta),$$
$$D_{\mathrm{SRKL}}^{(\alpha)}(\boldsymbol{x}, \boldsymbol{y}; p\|q_\theta) = D_{\mathrm{KL}}(\boldsymbol{x}, \boldsymbol{y}; q_\theta\|(1 - \alpha)p + \alpha q_\theta).$$

Despite the simple modification, SKL demonstrated higher convergence speed and achieved better performance compared to recent baselines, such as MiniLLM (Gu et al., 2024) and GKD (Agarwal et al., 2024). This effectiveness has been proven from both empirical and theoretical perspectives. For brevity, we will denote $D_{\mathrm{SKL}}^{(\alpha)}(\boldsymbol{x}, \boldsymbol{y}; p\|q_\theta)$ and $D_{\mathrm{SRKL}}^{(\alpha)}(\boldsymbol{x}, \boldsymbol{y}; p\|q_\theta)$ as $D_{\mathrm{SKL}}^{(\alpha)}(\boldsymbol{x}, \boldsymbol{y})$ and $D_{\mathrm{SRKL}}^{(\alpha)}(\boldsymbol{x}, \boldsymbol{y})$, respectively.

**Data curation of KD in LLMs.** To address the training inefficiency and low quality of SGO, which can lead to inaccurate teacher feedback in on-policy approaches (Lin et al., 2020; Agarwal et al., 2024), Ko et al. (2024) introduced an adaptive off-policy approach, which bridges offline and purely on-policy setups, striking a balance between the efficiency and efficacy of KD. This balanced strategy reuses SGO by introducing replay buffer, significantly improving computational efficiency while preserving the effectiveness of on-policy distillation. This approach has proven effective in subsequent works on preference alignment of LLMs (Rosset et al., 2024) as in more generalized version.

**Summary & Connection to our work.** Building on the insights from DistiLLM (Ko et al., 2024) – where SKL (or SRKL) and adaptive off-policy have shown efficacy – we introduce a contrastive approach that further refines these objectives. On the data curation side, we adopt a batch approach (Rosset et al., 2024) that collects SGO ahead of every training epoch in our setup, rather than on-policy approach, which samples at every training iteration. This also ensures compatibility with advanced LLM inference techniques, such as vLLM (Kwon et al., 2023), thereby

increasing generation efficiency and preserving the core philosophy of the adaptive off-policy approach. As shown in our preliminary results in Appendix D.1, this greatly reduces the computational cost of gathering training samples with minimal impact on student performance.

## 3. Method: DISTILLM-2

We introduce DISTILLM-2, a novel approach to LLM distillation, which lies in its new loss function as presented in Equation 2. This equips with a contrastive schema simultaneously accounting for different types of training responses (§3.1), along with dedicated data curation (§3.2) and curriculum-based adaptive loss design (§3.3).

$$\mathcal{L}_{\text{DISTILLM-2}} := \tag{2}$$
$$\frac{1}{2|\mathcal{D}|} \sum_{(\boldsymbol{x}, \boldsymbol{y}_t, \boldsymbol{y}_s) \sim \mathcal{D}} \left[ (1 - \beta) D_{\text{SKL}}^{(\alpha_t)}(\boldsymbol{x}, \boldsymbol{y}_t) + \beta D_{\text{SRKL}}^{(\alpha_s)}(\boldsymbol{x}, \boldsymbol{y}_s) \right],$$

where $D_{\text{SKL}}^{(\alpha_t)}(\boldsymbol{x}, \boldsymbol{y}_t)$ and $D_{\text{SRKL}}^{(\alpha_s)}(\boldsymbol{x}, \boldsymbol{y}_s)$ are SKL and SRKL tailored for teacher- and student-generated responses, respectively; $\mathcal{D}$ is the training dataset; $\beta$ is a coefficient in $[0, 1]$ to balance SKL and SRKL terms. In the following subsections, we provide detailed motivations, derivations, and use of the loss function in Equation 2 to formulate our DISTILLM-2 training process as stated in Algorithm 1.

### 3.1. Contrastive Approach

#### 3.1.1. MOTIVATION

**Concept.** Recently, contrastive approach in preference alignment, including DPO (Rafailov et al., 2023), which increases the likelihood of the preferred response ($\boldsymbol{y}_w$) while decreasing the likelihood of the dis-preferred response ($\boldsymbol{y}_l$), has demonstrated effective in enhancing LM performance.

$$-\log \sigma \underbrace{\left( \lambda \log \frac{q_\theta(\boldsymbol{y}_w|\boldsymbol{x})}{q_{\text{ref}}(\boldsymbol{y}_w|\boldsymbol{x})}}_{\text{increase } q_\theta(\boldsymbol{y}_w|\boldsymbol{x})} - \underbrace{\lambda \log \frac{q_\theta(\boldsymbol{y}_l|\boldsymbol{x})}{q_{\text{ref}}(\boldsymbol{y}_l|\boldsymbol{x})} \right)}_{\text{decrease } q_\theta(\boldsymbol{y}_l|\boldsymbol{x})}, \tag{3}$$

where $\sigma$ is sigmoid function, $q_{\text{ref}}$ is a reference model, and $\lambda$ is hyperparameter for DPO. This improvement stems from its dual mechanism: not only does it reduce the likelihood of undesired responses (Tajwar et al., 2024) but it also increases the likelihood of preferred responses, effectively reinforcing alignment with the desired behavior.

Similarly, we can apply this concept into KD to increase the likelihood of $q_\theta(\boldsymbol{y}_t|\boldsymbol{x})$ as match that of $p(\boldsymbol{y}_t|\boldsymbol{x})$ and decrease the likelihood of $q_\theta(\boldsymbol{y}_s|\boldsymbol{x})$ as match that of $p(\boldsymbol{y}_s|\boldsymbol{x})$ by bringing different types of loss function for each type of response. This approach allows better alignment of TGOs and SGOs in the contrastive manner than simply using a single type of loss function.

**Challenges of contrastive approach into KD.** While the concept itself is appealing, there are critical issues in di-

---

**Algorithm 1** Training pipeline of DISTILLM-2

1: **Input:** training iterations $T$, initial skew coefficient $\alpha_0$, teacher $p$, student $q_{\theta_0}$ with parameter $\theta_0$, prompt set
2: **Output:** Student model $q_{\theta_E}$ with trained parameters $\theta_E$
3: **for** epoch $e = 1, 2, \ldots, E$ **do**
4:     */* Sample *batched on-policy* responses */*
5:     **Sample** responses $\boldsymbol{y}_t, \boldsymbol{y}_s$ from teacher $p(\cdot|\boldsymbol{x})$ and student $q_{\theta_{e-1}}(\cdot|\boldsymbol{x})$ for given prompt $\boldsymbol{x}$
6:     **Construct** $\mathcal{D}_t = \{(\boldsymbol{x}, \boldsymbol{y}_t, \boldsymbol{y}_s)\}$ for training dataset for training epoch $e$.
7:     **Initialize** $\theta_e \leftarrow \theta_{e-1}$
8:     **for** iteration $\tau = 1, 2, \ldots, T$ **do**
9:         **Sample mini-batch:** $\mathcal{B} = \{(\boldsymbol{x}^{(i)}, \boldsymbol{y}_t^{(i)}, \boldsymbol{y}_s^{(i)})\}_{i=1}^{|\mathcal{B}|}$ from $\mathcal{D}_t$
10:         */* Curriculum-based adaptive update for $\alpha$ */*
11:         **Update** $\alpha_t \leftarrow 1 - (1 - \alpha_0) \cdot \frac{m}{p(\boldsymbol{y}_s|\boldsymbol{x}) - q_\theta(\boldsymbol{y}_s|\boldsymbol{x})}$ and $\alpha_s \leftarrow 1 - (1 - \alpha_0) \cdot \frac{m}{p(\boldsymbol{y}_t|\boldsymbol{x}) - q_\theta(\boldsymbol{y}_t|\boldsymbol{x})}$
12:         */* Gradual increasing coefficient for SRKL */*
13:         **Update** $\beta \leftarrow \texttt{clip}(\frac{e}{E} + \frac{\tau}{T}, \beta_0, 1)$
14:         */* Improved contrastive loss function (§3.3)*/*
15:         **Update** $\theta_e$ by minimizing $\mathcal{L}_{\text{DISTILLM-2}} = \frac{1}{2B} \sum \left[ (1 - \beta) D_{\text{SKL}}^{(\alpha_t)}(\boldsymbol{x}, \boldsymbol{y}_t) + \beta D_{\text{SRKL}}^{(\alpha_s)}(\boldsymbol{x}, \boldsymbol{y}_s) \right]$
16:     **end for**
17: **end for**

---

rectly applying DPO into KD. We observed that DPKD (Li et al., 2024b), which simply applies DPO by substituting the reference model with the teacher model, frequently suffers from reward hacking, leading to degenerate sentences:

$$-\log \sigma \underbrace{\left( \lambda \log \frac{q_\theta(\boldsymbol{y}_t|\boldsymbol{x})}{p(\boldsymbol{y}_t|\boldsymbol{x})} - \lambda \log \frac{q_\theta(\boldsymbol{y}_s|\boldsymbol{x})}{p(\boldsymbol{y}_s|\boldsymbol{x})} \right)}_{\text{inherently small } p(\boldsymbol{y}_s|\boldsymbol{x}) \to \text{overly decrease } q_\theta(\boldsymbol{y}_s|\boldsymbol{x})}, \tag{4}$$

where $\boldsymbol{y}_t$ and $\boldsymbol{y}_s$ are TGO and SGO, respectively. This is because DPKD only focuses on maximizing the gap between $\frac{q_\theta(\boldsymbol{y}_t|\boldsymbol{x})}{p(\boldsymbol{y}_t|\boldsymbol{x})}$ and $\frac{q_\theta(\boldsymbol{y}_s|\boldsymbol{x})}{p(\boldsymbol{y}_s|\boldsymbol{x})}$. As illustrated in Figure 1(b), we observe that this loss dynamics excessively decreases the likelihood of $q_\theta(\boldsymbol{y}_s|\boldsymbol{x})$ (*e.g.*, 91.25 in terms of negative log-likelihood; NLL), causing the student model to lose pre-trained information instead of fitting to teacher responses (*e.g.*, 20.29 in terms of NLL), as it replaces $q_{\text{ref}}$ with $p$ where $p(\boldsymbol{y}_s|\boldsymbol{x})$ is inherently small. Addressing this limitation requires rethinking and redesigning algorithm to integrate contrastive strategies into LLM distillation.

#### 3.1.2. CONTRASTIVE APPROACH FOR LLM DISTILLATION

To bring contrastive strategy into KD, we propose a new loss function $\mathcal{L}_{\text{CALD}}$, using a combination of SKL and SRKL (Ko et al., 2024). Our design stems from the follows.

**Observation on behavior of KL and RKL.** Here, we pro-

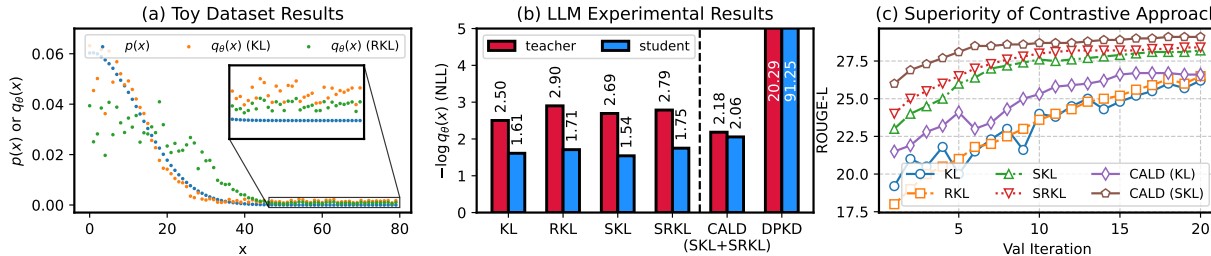

*Figure 1.* (a) The behavior of KL (orange) and RKL (green) is analyzed for long-tailed toy data introduced in Wu et al. (2024). (b) NLL of student models on teacher (red) and student (blue) responses, using Mistral-7B and Danube2-1.8B as the teacher and student models, respectively, optimized with diverse loss functions. (c) We propose CALD with SKL and SRKL that achieves faster convergence and higher ROUGE-L (Lin, 2004), following the experimental setup of Ko et al. (2024). Detailed setup can be found in Appendix D.1.

vide an observation on the behavior of KL and RKL: they can increase and decrease the likelihood of $q_\theta$ for TGOs (KL) and SGOs (RKL), respectively. As shown in Figure 1, KL increases $q_\theta(\cdot|\boldsymbol{x})$ in regions where $p(\cdot|\boldsymbol{x})$ are high (*i.e.*, **pulling-up effect**). For example, this occurs in the head of the teacher distribution in Figure 1(a) or for TGOs in Figure 1(b) – This behavior arises because they aim to focus on reducing the ratio $\frac{p(\cdot|\boldsymbol{x})}{q_\theta(\cdot|\boldsymbol{x})}$ for the region where $p(\cdot|\boldsymbol{x})$ are large to minimize weighted average. Conversely, RKL attempts to reduce the ratio $\frac{q_\theta(\cdot|\boldsymbol{x})}{p(\cdot|\boldsymbol{x})}$. Consequently, $q_\theta(\cdot|\boldsymbol{x})$ decreases in region where $p(\cdot|\boldsymbol{x})$ are small (*i.e.*, **pushing-down effect**), such as the tail of teacher distribution in Figure 1(a) or student responses in Figure 1(b). Detailed mathematical explanation can be found in Appendix B.1.

**Our solution.** For implementing CALD, an optimal choice among various KL-based loss functions would be one that demonstrates state-of-the-art results while exhibiting similar behavior to KL and RKL, as observed in Figure 1. To this end, we utilize skew KL (SKL) and RKL (SRKL), introduced in DistiLLM (Ko et al., 2024), as the backbone loss functions. Specifically, we design the loss function for CALD, using SKL for teacher responses (*i.e.*, $\boldsymbol{y}_t$) where most of $p(\boldsymbol{y}_t|\boldsymbol{x}) \gg 0$ and using SRKL for student responses, $\boldsymbol{y}_s$, where the most of $p(\boldsymbol{y}_s|\boldsymbol{x}) \simeq 0$. Formally, our proposed loss function can be written as follows:

$$\mathcal{L}_{\text{CALD}} = \frac{1}{2|\mathcal{D}|} \sum_{(\boldsymbol{x},\boldsymbol{y}_t,\boldsymbol{y}_s) \sim \mathcal{D}} D^{(\alpha)}_{\text{SKL}}(\boldsymbol{x},\boldsymbol{y}_t) + D^{(\alpha)}_{\text{SRKL}}(\boldsymbol{x},\boldsymbol{y}_s).$$
(5)

Despite its simplicity, **this loss function implies that the importance of simultaneous consideration of responses type during objective function design**. Note that Ko et al. (2024) demonstrated that a vanilla interpolation between $\gamma D^{(\alpha)}_{\text{SKL}}(\boldsymbol{x},\cdot) + (1-\gamma)D^{(\alpha)}_{\text{SRKL}}(\boldsymbol{x},\cdot)$ for all $\gamma \in [0,1]$ over the same type of responses, (*e.g.*, either $\boldsymbol{y}_t$ or $\boldsymbol{y}_s$), does not improve performance compared to using either SKL or SRKL alone. However, we find that the new approach of using different types of responses for different terms significantly enhances performance. $\mathcal{L}_{\text{CALD}}$ achieves faster convergence and greater effectiveness compared to the ex-

clusive use of SKL or SRKL in DistiLLM (see Figure 1(c)). Note that while simple KL and RKL also prove effectiveness for CALD, using SKL and SRKL as backbone achieves higher efficacy, consistent with Ko et al. (2024).

**Mathematical connection to DPKD and DPO.** We now reveal that our proposed loss function $\mathcal{L}_{\text{CALD}}$ can be mathematically interpreted as exhibiting similar *yet different* behavior to DPKD (or DPO).

**Remark 1.** *Equation 5 can be re-written as follows:*

$$-\mathbb{E}_{\substack{\boldsymbol{y}_t \sim p(\cdot|\boldsymbol{x}), \\ \boldsymbol{y}_s \sim q_\theta(\cdot|\boldsymbol{x})}} \left[ \frac{1}{\lambda} \cdot \left( \lambda \log \frac{\tilde{q}_\theta(\boldsymbol{y}_t|\boldsymbol{x})}{p(\boldsymbol{y}_t|\boldsymbol{x})} - \lambda \log \frac{q_\theta(\boldsymbol{y}_s|\boldsymbol{x})}{\tilde{p}(\boldsymbol{y}_s|\boldsymbol{x})} \right) \right],$$
(6)

*where* $\tilde{q}_\theta(\cdot|\boldsymbol{x}) = \alpha p(\cdot|\boldsymbol{x}) + (1-\alpha)q_\theta(\cdot|\boldsymbol{x})$ *and* $\tilde{p}(\cdot|\boldsymbol{x}) = \alpha q_\theta(\cdot|\boldsymbol{x}) + (1-\alpha)p(\cdot|\boldsymbol{x})$.

This indicates CALD enable to increase $\tilde{q}_\theta(\boldsymbol{y}_t|\boldsymbol{x})$ (and implicitly $q_\theta(\boldsymbol{y}_t|\boldsymbol{x})$) and decrease $q_\theta(\boldsymbol{y}_s|\boldsymbol{x})$, simultaneously. The detailed derivation can be found in Appendix B.2.

Despite this similarity, there are two critical and non-trivial differences between CALD and DPKD (or DPO). *First*, rather than employing the log-sigmoid function used in DPKD, Equation 6 adopts a linear formulation that allows token-level decomposition and explicit weighting by $p(\boldsymbol{y}_t|\boldsymbol{x})$ or $q_\theta(\boldsymbol{y}_s|\boldsymbol{x})$ (as in Equation 1). *Second*, by inherently linear dependency between $\tilde{q}_\theta(\cdot|\boldsymbol{x})$ and $p(\cdot|\boldsymbol{x})$ (or between $\tilde{p}(\cdot|\boldsymbol{x})$ and $q_\theta(\cdot|\boldsymbol{x})$), this regularizes the overly decreasing $q_\theta(\boldsymbol{y}_s|\boldsymbol{x})$, which resolves the challenges in DPKD. From this, CALD (*i.e.,* DISTILLM-2) outperforms DPO and DPKD by a large margin, as shown in Appendix D.1.

### 3.2. Optimal Data Curation for Contrastive Approach

In the context of datasets for LLM distillation, one common question might be:

> *"How can we effectively utilize well given SGO and high-quality fixed datasets in distillation of LLMs?"*

While previous works (Xu et al., 2024b; Li et al., 2024a) have proposed effective strategies for leveraging these two complementary dataset types in an SFT manner, we observed that their techniques – such as speculative generation

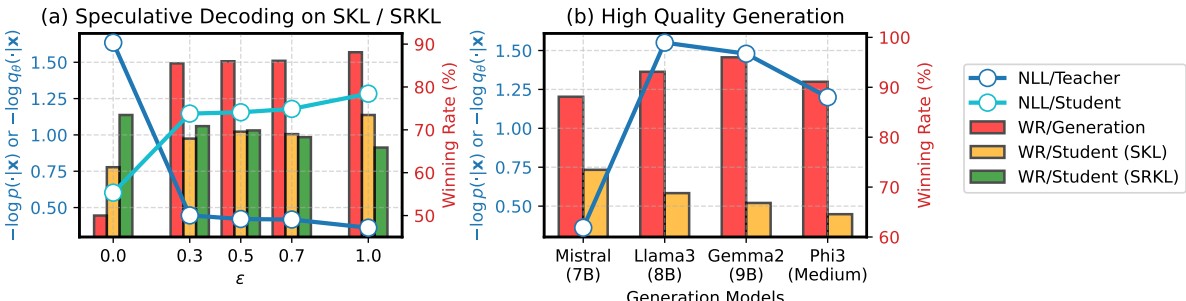

*Figure 2.* Comparison of the winning rates compared to the student before KD (**WR**) of student models with (a) replacing $\boldsymbol{y}_t$ (orange) or $\boldsymbol{y}_s$ (green) with $\boldsymbol{y}_{\text{spec}}$, responses from speculative decoding (Cai et al., 2024), varying the hyperparameter $\varepsilon$. (b) replacing $\boldsymbol{y}_t$ with responses generated using stronger LLMs (e.g., Llama-3, Gemma-2, Phi-3) than the teacher models (i.e., Mistral) for the SKL term. We also show the negative log-likelihood (NLL) of the student (cyan) and teacher (blue) models on the replaced responses, along with the corresponding WR (red).

*Table 1.* Motivation for curriculum approach for $\alpha$. UF and EI indicate the winning rates (%) of responses on the UltraFeedback and Evol-Instruct test sets, compared to the student model before iteration 1, as judged by GPT-4o-mini.

| | | Epoch. 1 | | | Epoch. 2 | | | Epoch. 3 | |
|---|---|---|---|---|---|---|---|---|---|
| | $\alpha$ | UF (%) | EI (%) | $\alpha$ | UF (%) | EI (%) | $\alpha$ | UF (%) | EI (%) |
| | 0.3 | 71.53 | 76.23 | - | - | - | - | - | - |
| | | | | 0.1 | 72.18 | 81.18 | - | - | - |
| **Mistral** | 0.1 | **73.46** | **79.83** | | | | 0.1 | 73.25 | 82.86 |
| | | | | 0.01 | **75.75** | **84.35** | 0.01 | **76.59** | **84.67** |
| | 0.01 | 70.35 | 76.38 | - | - | - | - | - | - |
| | 0.3 | 69.89 | 75.86 | - | - | - | - | - | - |
| **Qwen2** | 0.1 | 70.01 | **76.21** | 0.1 | **74.30** | **81.23** | 0.1 | **75.32** | **81.81** |
| | | | | 0.01 | 69.95 | 78.86 | - | - | - |
| | 0.01 | 66.62 | 74.82 | - | - | - | - | - | - |

Entries marked "-" were omitted as they were found sub-optimal in previous epochs.

or the use of high-quality responses (which may outperform teacher generations) – are less effective in CALD. From our further discussion, **we conclude that utilizing teacher and student generations for SKL and SRKL, respectively, may be the optimal strategy for CALD**, as it consistently aligns with the core philosophy of CALD.

**Exploring the trade-offs between teacher and student generations.** Previous works (Agarwal et al., 2024) have discussed that while teacher responses provide useful information, they can cause training-inference mismatches. In contrast, SGOs, though lower in quality, effectively reduce such mismatches, leading to higher efficacy. To explore these complementary perspectives, we use speculative decoding[12] to find the key factors for dataset curation.

In speculative generation, student drafts $K$ tokens, and teacher verify them in parallel for $1 \leq k \leq K$ based on[3]:

$$q_\theta(y_{n+k}|\boldsymbol{y}_{<n+k}) > \min(\varepsilon^2, \varepsilon \cdot \exp(-H(p(\cdot|\boldsymbol{y}_{<n+k})))),$$

where $H(\cdot)$ and $\varepsilon$ are entropy function and hyperparameter.

When we replace $\boldsymbol{y}_t$ with $\boldsymbol{y}_{\text{spec}}$, as $\varepsilon$ decreases (*i.e.,* more acceptance of drafts), the distilled model better aligns with the student distribution $q_\theta(\cdot|\boldsymbol{x})$. However, as shown with orange bars, its performance is highest with $\boldsymbol{y}_{\text{spec}}$ at $\varepsilon = 1.0$ (*i.e.,* identical to $\boldsymbol{y}_t$). This implies that on the SKL side, mitigating the training-inference mismatch via SGOs does not always lead to performance improvement. Rather, strong guidance from the teacher response is highly related to the distillation performance.

Conversely, on the responses for SRKL, the distilled model achieves the highest performance with $\boldsymbol{y}_{\text{spec}}$ at $\varepsilon = 0.0$ (*i.e.,* identical to $\boldsymbol{y}_s$), as shown with green bars, although these responses are of the lowest quality. This implies that using low-quality SGO samples on the SRKL side may be beneficial for our contrastive approach. The effectiveness of reduced training-inference mismatch via SGOs can be attributed to this edge of alignments.

**High-quality does not always guarantee success.** One additional question that arises is whether the success of teacher responses on the SKL term is due to their higher quality. It is natural to consider if using higher-quality responses from powerful LLMs like ChatGPT would improve performance, similar to black-box KD (Li et al., 2024a). To investigate, we replaced the responses for SKL term with those generated from stronger LLMs (*e.g.*, Llama3-8B) instead of the Mistral-7B teacher's responses. As shown in Figure 2(b), although these stronger LLMs generate high-quality answers, the student trained on the teacher's responses still performs

---

[1]The original work primarily aims to accelerate generation without sacrificing the quality of generated responses. If we use $\boldsymbol{y}_{\text{spec}}$ instead of $\boldsymbol{y}_t$ for SKL, we may mitigate training-inference mismatch, potentially improving overall performance by aligning the training data more closely with student distribution.

[2]Alternatively, $\boldsymbol{y}_{\text{spec}}$ for SRKL could improve student performance by training on higher-quality samples. However, as discussed in this section, this approach did not yield the desired results.

[3]Specifically, we applied speculative decoding with typical decoding (Cai et al., 2024), as it simplifies interpolation compared to rejection sampling-based methods (Leviathan et al., 2023).

*Table 2.* Comparison winning rates (WR) using pairwise comparison (Zheng et al., 2023) on three instruction-following benchmarks. The baseline is `text-davinci-003` in AlpacaEval and `gpt-3.5-turbo` in Evol-Instruct and UltraFeedback. The judges are GPT-4o for AlpacaEval and Evol-Instruct, GPT-4o-mini for UltraFeedback. The best and the second best win rates are in **bold** and underline.

| Method | Qwen2-7B-Inst ($\mathcal{M}_T$) → Qwen2-1.5B ($\mathcal{M}_S$) | | | | Mistral-7B-Inst ($\mathcal{M}_T$) → Danube2-1.8B ($\mathcal{M}_S$) | | | | Gemma-2-9B-Inst ($\mathcal{M}_T$) → Gemma-2-2B ($\mathcal{M}_S$) | | | |
|---|---|---|---|---|---|---|---|---|---|---|---|---|
| | AlpacaEval WR(%) | Evol-Inst WR(%) | UltraFeed WR(%) | AVG. WR(%) | AlpacaEval WR(%) | Evol-Inst WR(%) | UltraFeed WR(%) | AVG. WR(%) | AlpacaEval WR(%) | Evol-Inst WR(%) | UltraFeed WR(%) | AVG. WR(%) |
| $\mathcal{M}_T$ | 88.41 | 70.70 | 69.25 | 76.12 | 91.92 | 73.51 | 83.59 | 83.01 | 95.78 | 88.76 | 85.90 | 90.15 |
| $\mathcal{M}_S$ | 51.06 | 18.00 | 21.93 | 30.33 | 48.17 | 12.84 | 20.06 | 27.02 | 42.51 | 16.74 | 26.60 | 28.62 |
| KD | 57.49 | 28.23 | 37.86 | 41.19 | 60.21 | 18.23 | 41.56 | 40.00 | 61.78 | 32.45 | 54.37 | 49.53 |
| SeqKD | 58.02 | 29.11 | 38.35 | 41.83 | 59.76 | 18.45 | 42.11 | 40.11 | 62.43 | 33.21 | 55.18 | 50.27 |
| ImitKD | 59.37 | 30.58 | 39.92 | 43.29 | 58.34 | 17.89 | 40.87 | 39.03 | 63.12 | 31.89 | 53.92 | 49.64 |
| GKD | 66.07 | 44.61 | 57.74 | 56.14 | 69.75 | 24.54 | 57.74 | 50.68 | 81.43 | 50.57 | 77.20 | 69.73 |
| DistiLLM | 66.30 | 44.61 | 58.18 | 56.35 | 70.16 | 28.78 | 58.18 | 52.37 | 82.95 | 51.26 | 76.68 | 70.30 |
| Speculative KD | 61.52 | 44.95 | 56.82 | 54.43 | 64.58 | **38.87** | 60.04 | 54.50 | 78.45 | 57.11 | 72.21 | 69.26 |
| **DISTILLM-2** | **69.88** | **47.13** | **59.05** | **58.69** | **74.04** | 32.84 | **62.46** | **56.45** | **85.97** | **59.53** | **78.99** | **74.83** |

better. This suggests that the high log-probability of responses from the teacher model may be a more important factor in data curation than their higher quality.

**Discussion on the observations.** These findings align with the motivation of CALD in §3.1: the "pulling-up" effect of SKL is maximized at the head of $p(\cdot|\boldsymbol{x})$ (i.e., $\boldsymbol{y}_t$), while the "pushing-down" effect of SRKL is maximized at the tail of $p(\cdot|\boldsymbol{x})$ (i.e., $\boldsymbol{y}_s$). ***First***, while speculative generations are effective with vanilla KL in Speculative KD (Xu et al., 2024b), they are less effective with our contrastive loss because (1) speculative generations are an interpolation of $\boldsymbol{y}_t$ and $\boldsymbol{y}_s$, which may weaken both the "pulling-up" and "pushing-down" effect – core mechanisms underlying CALD; and (2) the contrastive loss already exploits both complementary response types simultaneously, reducing the need for interpolation compared to single-loss settings. ***The second observation*** also supports our claim that pure teacher generation may be optimal for SKL where they completely align with $p(\cdot|\boldsymbol{y})$, rather than relying on higher-quality responses, from the perspective of maximizing the "pulling-up" effect at the head of $p(\cdot|\boldsymbol{x})$.

### 3.3. Curriculum-based Adaptive Learning

We introduce two modifications, inspired by our empirical observations, to implement difficulty-based adaptive learning and facilitate the conversion from Equation 6 to Equation 2: a **curriculum approach for $\alpha$** and a **gradual increasing of coefficient for SRKL**.

**Curriculum Approach for $\alpha$.** One limitation of SKL (Ko et al., 2024) is that we need to manually determine $\alpha$, which interpolates between the teacher and student distributions. A larger $\alpha$ improves optimization stability and accelerates convergence, but it limits the acquisition of informative knowledge by inherently small gap between $p(\cdot)$ and $\alpha p(\cdot) + (1-\alpha)q_\theta(\cdot)$. Conversely, a smaller $\alpha$ allows for greater knowledge acquisition but reduces optimization stability and slows convergence (Ko et al., 2024). While previous work

suggests that $\alpha$ values in a moderate range (*e.g.*, 0.1–0.3) are generally robust, we observed that the optimal values can still vary across different setups due to the variation of teacher-student pairs and the dynamic requirements of different training epochs (see Table 1).

Regarding the dynamic of different training epoch, we observe that the optimal values for $\alpha$ for the second or third epoch are either equal to or smaller than than those in the first epoch (Table 1). Building on this observation, we propose a curriculum-based approach for updating $\alpha$. For "easy" samples, where $p(\cdot)$ and $q_\theta(\cdot)$ are sufficiently similar, we select a small $\alpha$. On the other hand, for "hard" samples, where the difference between $p(\cdot)$ and $q_\theta(\cdot)$ is large, we choose a larger $\alpha$.

To implement this, we introduce an updating rule for $\alpha \in [0, 1]$ based on the following approximation:

$$\log \frac{p(\boldsymbol{y}|\boldsymbol{x})}{\tilde{q}_\theta^{(\alpha)}(\boldsymbol{y}|\boldsymbol{x})} \simeq (1-\alpha) \cdot (p(\boldsymbol{y}|\boldsymbol{x}) - q_\theta(\boldsymbol{y}|\boldsymbol{x})), \quad (7)$$

where $\tilde{q}_\theta^{(\alpha)}(\boldsymbol{y}|\boldsymbol{x}) = \alpha p(\boldsymbol{y}|\boldsymbol{x}) + (1-\alpha)q_\theta(\boldsymbol{y}|\boldsymbol{x})$. Note that this approximation originates from the Mercator series expansion (Zwillinger, 2002): $\log(1+x) = \sum_{n=1}^{\infty}(-1)^{n+1} \cdot \left(\frac{x^n}{n}\right)$. This series allows the first-order approximation $\log p(x) \simeq p(x) - 1$. The detailed derivation can be found in Appendix. Using this formula, we can compute a suitable $\alpha$ in closed-form for each sample, allocating proper $\alpha$ by making $(1-\alpha) \cdot (p(\cdot) - q_\theta(\cdot))$ consistent across entire training. The detailed implementation for this updating rule can be found in Algorithm 1.

**(Linearly) Gradual increasing of coefficient for SRKL.** Based on the behavior of TGOs with SKL and SGOs with SRKL in Equation 5, the first term enables the acquisition of advanced information by matching high-probability on TGOs, while the second term suppresses undesirable behavior by preventing the matching of similarly low probabilities in SGOs. However, achieving $q_\theta(\cdot|\boldsymbol{x}) = p(\cdot|\boldsymbol{x})$ for all $\boldsymbol{y}_t$ and $\boldsymbol{y}_s$ is challenging with limited dataset sizes due to inher-

*Table 3.* Comparison results on the GSM8k and MATH benchmarks. The best *pass@1* score is highlighted in **bold**.

| Method | Qwen2-Math-7B-Inst ($\mathcal{M}_T$) → Qwen2-Math-1.5B ($\mathcal{M}_S$) | | | Qwen2.5-Math-7B-Inst ($\mathcal{M}_T$) → Qwen2.5-Math-1.5B ($\mathcal{M}_S$) | | |
|---|---|---|---|---|---|---|
| | GSM8K Pass@1 | MATH Pass@1 | AVG. Pass@1 | GSM8K Pass@1 | MATH Pass@1 | AVG. Pass@1 |
| $\mathcal{M}_T$ | 83.93 | 41.28 | 62.61 | 89.31 | 44.82 | 67.07 |
| $\mathcal{M}_S$ | 74.53 | 25.56 | 50.05 | 77.33 | 27.14 | 52.24 |
| GKD | 75.44 | 34.16 | 54.80 | 80.21 | 40.54 | 60.38 |
| DistiLLM | 75.59 | 34.54 | 55.07 | 81.05 | 41.14 | 61.10 |
| DISTILLM-2 | **76.27** | **35.58** | **55.93** | **81.20** | **42.94** | **62.07** |

*Table 4.* Comparison results on the HumanEval (HEval) and MBPP benchmarks. The best *pass@1* score is highlighted in **bold**.

| Method | DS-Coder-6.9B-Inst ($\mathcal{M}_T$) → DS-Coder-1.3B ($\mathcal{M}_S$) | | | Qwen2.5-Coder-7B-Inst ($\mathcal{M}_T$) → Qwen2.5-Coder-1.5B ($\mathcal{M}_S$) | | |
|---|---|---|---|---|---|---|
| | HEval Pass@1 | MBPP Pass@1 | AVG. Pass@1 | HEval Pass@1 | MBPP Pass@1 | AVG. Pass@1 |
| $\mathcal{M}_T$ | 85.37 | 82.54 | 83.96 | 75.61 | 74.60 | 75.61 |
| $\mathcal{M}_S$ | 50.61 | 72.22 | 61.42 | 30.73 | 60.84 | 45.79 |
| GKD | 54.88 | 74.34 | 64.61 | 40.85 | 61.90 | 51.38 |
| DistiLLM | 53.65 | 74.34 | 64.00 | 39.63 | 62.17 | 50.90 |
| DISTILLM-2 | **59.92** | **75.66** | **67.79** | **42.24** | **62.70** | **52.47** |

*Table 5.* Component analysis of DISTILLM-2 including (1) applying a contrastive loss, (2) increasing $\beta$, and (3) introducing curriculum-based updates to $\alpha$. When all components are applied to DistiLLM (v1), it becomes identical to DISTILLM-2.

| | (1) | (2) | (3) | Qwen2 (↑) | Danube2 (↑) | Gemma2 (↑) | AVG. (↑) |
|---|---|---|---|---|---|---|---|
| DistiLLM (v1) | | | | 44.61 | 28.78 | 51.26 | 41.55 |
| | ✓ | | | 45.41 | 30.73 | 54.70 | 43.61 |
| | ✓ | ✓ | | 45.87 | 31.88 | 56.65 | 44.80 |
| | ✓ | | ✓ | 46.33 | 31.65 | 57.68 | 45.22 |
| DISTILLM-2 | ✓ | ✓ | ✓ | **47.24** | **32.80** | **59.53** | **46.50** |

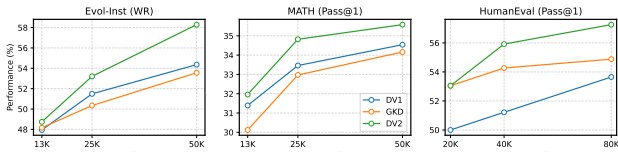

*Figure 3.* Comparison of performance for different sizes of training datasets across a wide range of tasks.

ent capacity gap between the teacher and student models, which arises from factors such as the number of parameters. Nevertheless, we observe that *gradually increasing the SRKL coefficient $\beta$ in* Equation 2, following the linear schedule shown in Algorithm 1, significantly improves student performance (see Table 5). This improvement is achieved by compromising the imitation of the teacher's behavior, which is relatively hard to achieve, while focusing on directly obtaining feedback from SGOs, thereby effectively reducing the training–inference mismatch.

## 4. Experiments

### 4.1. General Instruction-Following

**Setup.** We first construct the training datasets, by randomly sampling 50k prompts from UltraChat200k (Ding et al., 2023) and use the corresponding teacher and student to generate the responses. After training, we evaluate DISTILLM-2 for general purpose instruction-following task on AlpacaEval (Li et al., 2023), Evol-Instruct (Xu et al., 2024a), and UltraFeedback (Cui et al., 2024). For evaluation, we adopt LLM-as-a-Judge (Zheng et al., 2023) with GPT-4o or GPT-4o-mini as judge models. For experiments, we use Qwen2-7B (Hui et al., 2024), Mistral-7B (Jiang et al., 2023), Gemma-2-9B (DeepMind et al., 2024) instruction models as teachers and Qwen2-1.5B, Danube2-1.8B (Singer et al., 2024), and Gemma2-2B as students, respectively.

**Results.** We report experimental results in Table 2. This comparison between DISTILLM-2 and other baselines shows that our proposed method performs best in the most of evaluation setups, except for Danube2-1.8B in Evol-

Instruct. It outperforms the second best methods by **+2.34%**, **+1.95%**, and **+4.53%** on average for Qwen2-1.5B, Danube2-1.8B, and Gemma2-2B, respectively. As these evaluation benchmarks cover a wide range of domains relevant to real-world applications, these results demonstrate that DISTILLM-2 can be widely used to build strong sLMs.

### 4.2. Mathematical Reasoning

**Setup.** We conduct experiments on two standard mathematical reasoning benchmarks: GSM8K (Cobbe et al., 2021) and MATH (Hendrycks et al., 2021). For teacher and student pairs, we select the Qwen2-Math-7B-Inst and Qwen2.5-Math-7B-Inst as teacher models and Qwen2-Math-1.5B and Qwen2.5-Math-1.5B as student models, respectively. The student models are trained using 50k randomly selected samples from the MetaMathQA (Yu et al., 2024a) dataset. Specifically, the student models are fine-tuned in a supervised manner on the entire MetaMathQA for a single epoch.

**Results.** Table 3 summarizes the effectiveness of DISTILLM-2 compared to recent competitive baselines, including GKD and DistiLLM. In both the Qwen2 and Qwen2.5 experimental setups, DISTILLM-2 achieves higher performance than other baselines on the GSM8K and MATH evaluations. Interestingly, the Qwen2.5 student with DISTILLM-2 demonstrates competitive average performance and even outperforms the Qwen2 teacher on the MATH evaluation, which is a challenging milestone.

### 4.3. Code Generation

**Setup.** We utilize prompts from WizardCoder (Luo et al., 2024) dataset which is developed using the Evol-Instruct method (Xu et al., 2024a) code instruction datasets. We apply Qwen2.5-Coder-7B-Inst (Hui et al., 2024) and

*Table 6.* Comparison of preference optimization results (WR %) using different reference models from various KD methods.

| Method | Qwen2-1.5B-SFT ($\mathcal{M}_S$) | | | Gemma-2-2B-SFT ($\mathcal{M}_S$) | | |
|---|---|---|---|---|---|---|
| | Evol | Ultra | AVG. | Evol | Ultra | AVG. |
| $\mathcal{M}_S$ | 26.92 | 33.25 | 30.09 | 41.34 | 50.85 | 46.10 |
| GKD | 47.84 | 65.48 | 56.66 | 59.13 | 82.83 | 70.98 |
| DistiLLM | 48.32 | 65.18 | 56.75 | 57.21 | 79.95 | 68.58 |
| DISTILLM-2 | 52.88 | 67.78 | 60.33 | 68.99 | 83.58 | 76.29 |

DeepSeek-Coder-6.7B-Inst as teacher models and Qwen2.5-Coder-1.5B and DeepSeek-Coder-1.3B as student models, respectively. Similarly, we train the student models for 2 epochs. We evaluate performance on two standard coding benchmarks: HumanEval (Chen et al., 2021) and MBPP (Austin et al., 2021).

**Results.** The results are presented in Table 4. Across both HumanEval and MBPP, DISTILLM-2 consistently outperforms the baseline methods, GKD and DistiLLM. Notably, GKD achieves higher scores than DistiLLM, its effectiveness remains lower than that of DISTILLM-2. This outcome highlights DISTILLM-2 's ability to integrate a specialized alignment strategy – effectively incorporating SKL (or SRKL) with harmonized response type.

## 5. Additional Ablation Study

Here, we provide additional ablation experiments on DISTILLM-2. Our ablation studies are conducted with Qwen2 (or Qwen2.5) from its diversified model sizes. We use GPT-4o-mini as a judge model for all ablation studies due to its cost-efficiency.

**Component Analysis.** Here, we conducted a component analysis of DISTILLM-2 's technical components, which included (1) applying a contrastive approach (§3.1 & §3.2), (2) increasing the $\beta$ parameter, and (3) introducing curriculum-based updates to $\alpha$ (§3.3). Table 5 shows a component-wise analysis demonstrating how progressively incorporating these improvements into DistiLLM brings its performance in line with that of DISTILLM-2. As each component is added, we observe incremental performance gains, indicating that all of the examined components enhance DISTILLM-2 's overall effectiveness.

**Training Size.** We investigated how varying the training data size affects the performance of DISTILLM-2. In Figure 3, we show its performance across various tasks—such as instruction-following, math reasoning, and code generation—and compare it against baselines including GKD and DistiLLM. We observe that our proposed method consistently outperforms these baselines, demonstrating the highest effectiveness among all considered LLM distillation methods.

**Capacity Gap.** It is well known that a substantial capac-

*Table 7.* Evaluation on OK-VQA and TextVQA, two popular benchmark for visual question answering. We utilized VQA accuracy (Antol et al., 2015). The best results are highlighted in **bold**.

| VQA Acc. (%) | $\mathcal{M}_T$ | $\mathcal{M}_S$ | GKD | DistiLLM | DISTILLM-2 |
|---|---|---|---|---|---|
| **OK-VQA** | 54.70 | 36.87 | 41.83 | 39.38 | **44.72** |
| **TextVQA** | 42.91 | 28.34 | 33.84 | 31.10 | **34.98** |
| **AVG.** | 48.81 | 32.61 | 37.84 | 35.24 | **39.85** |

*Table 8.* Performance of the Qwen1.5 series with a 0.5B student model and varying teacher. The last column ($\Delta$) shows the improvement over Ko et al. (2024). The performance metric is the winning rate compared to the original student models on Evol-Instruct.

| Size | $\mathcal{M}_T$ | GKD | DistiLLM | DISTILLM-2 | $\Delta$ |
|---|---|---|---|---|---|
| 1.8B ($\uparrow$) | 81.07 | 64.18 | 65.23 | **66.31** | +1.08 |
| 7B ($\uparrow$) | 92.71 | 71.15 | 72.11 | **74.86** | +2.75 |
| 14B ($\uparrow$) | 95.16 | 72.59 | 72.11 | **76.78** | +4.67 |

ity gap between large teacher models and compact student models makes KD more challenging, a phenomenon referred to as the capacity gap (Mirzadeh et al., 2020). We experimented with diverse size of Qwen-1.5-Chat with 1.8B, 7B, and 14B parameters as teacher and SFT of Qwen1.5-0.5B student to seize the behavior of DISTILLM-2 across the different size of teacher models. As shown in Table 8, DISTILLM-2 demonstrates monotonic improvement and consistently outperforms other baselines as the teacher size increases. This result highlights DISTILLM-2's effectiveness in addressing capacity gap issues, whereas the previous version (Ko et al., 2024) struggled with capacity gaps, particularly with the 7B and 14B teacher models.

## 6. Broader Impacts

Furthermore, we present a range of diverse applications for DISTILLM-2, demonstrating its broad versatility and highlighting its potential for future use. We also provide additional applications of DISTILLM-2 in Appendix 6.3 (*i.e.,* recovering quantized model) and 6.4 (*i.e.,* achieving higher inference speed in speculative decoding).

### 6.1. Additional Results for DISTILLM-2 + DPO

In preference alignment (Ouyang et al., 2022; Rafailov et al., 2023), training usually involves two steps: (1) SFT and (2) preference fine-tuning using either PPO or DPO. While most previous works have concentrated on the second step, we highlight that the first step is also important. In our study, we replace the standard SFT method with KD of LLMs and evaluate its effectiveness by comparing how the policy LLMs perform after the second step, which uses DPO. Specifically, we use the reference model for each setup as trained student in Table 2. The results in Table 6 show that replacing SFT with distillation in the first phase leads to

*Table 9.* Performance of Phi-3.5-mini-instruct at different levels of precision. The best results are highlighted in **bold**.

| Size | AlpacaEval | Evol-Inst | UltraFeed | AVG. |
|---|---|---|---|---|
| $\mathcal{M}_T$ (BFloat16) | 91.80 | 82.80 | 80.54 | 85.05 |
| $\mathcal{M}_S$ (INT4) | 84.32 | 74.89 | 77.60 | 78.94 |
| GKD (INT4) | 88.57 | 80.04 | **79.80** | 82.80 |
| DistiLLM (INT4) | 89.13 | 81.77 | 79.15 | 83.35 |
| DISTILLM-2 (INT4) | **89.17** | **81.96** | 79.58 | **83.57** |

*Table 10.* Comparison of the inference speedup of speculative decoding using different draft models obtained from various KD methods for the target models Phi3-medium and Phi3.5-mini.

| Phi- | | | SFT | GKD | DistiLLM | DISTILLM-2 |
|---|---|---|---|---|---|---|
| 3-medium | Spd. (↑) | | ×1.32 | ×1.64 | ×1.71 | **×1.97** |
| | Acpt. (↑) | | 0.412 | 0.464 | 0.469 | **0.487** |
| 3.5-mini | Spd. (↑) | | ×1.24 | ×1.58 | ×1.65 | **×1.84** |
| | Acpt. (↑) | | 0.397 | 0.443 | 0.452 | **0.522** |

higher overall alignment performance in preference fine-tuning. Notably, our DISTILLM-2 achieved a substantially higher WR, more than doubling the WR in Qwen2-1.5B, and showed a similar improvement in Gemma2-2B. These indicate that DISTILLM-2 can build effective reference models for the subsequent preference alignment phase.

### 6.2. Expansion to Vision-Language Models

We also applied DISTILLM-2 on the distillation setup of vision-language models (VLMs) to boost the versatility of proposed method that can be applied in a wide range of modalities. We select LLaVA-1.5-7B (Liu et al., 2024) and TinyLLaVA-1.4B (Zhou et al., 2024a) as a teacher and a student model, respectively. For training dataset, we utilize the prompt from RLAIF-V-Dataset (Yu et al., 2024b) which contains 83K prompts, and evaluate the trained models on two popular benchmark, OK-VQA (Marino et al., 2019) and TextVQA (Singh et al., 2019). Table 7 shows that the superiority of DISTILLM-2 over other distillation methods holds true not only in LLM setups but also with VLMs. Although other baselines also demonstrated effectiveness compared to original student models (i.e., $\mathcal{M}_S$), DISTILLM-2 outperformed GKD and DistiLLM by +2.01% and +4.61% on average, respectively.

### 6.3. Restoring the Performance of Quantized LLMs

Using parameter-efficient fine-tuning methods, such as LoRA, can help recover the performance of quantized LLMs after post-training quantization (Frantar et al., 2023), introducing only a negligible number of additional parameters. Here, we demonstrate the effectiveness of DISTILLM-2 in restoring the performance of 4-bit quantized LLMs using LoRA by replacing regular SFT with KD baselines. Figure 9 shows that all KD methods can significantly improve the performance of quantized models while adding only a few trainable parameters. Additionally, DISTILLM-2 achieves the best average performance among KD baselines. Distillation can also be straightforwardly applied to pairs of original and compressed models, such as pruned or quantized versions, enabling efficient deployment on mobile devices.

### 6.4. Inference Speedup of Speculative Decoding

DistillSpec (Zhou et al., 2024b) demonstrate that KD can improve speculative decoding by better aligning the drafter

and verifier models. Building on their work, we evaluate the inference speedup of speculative decoding using Phi3.5-mini and Phi3-medium (Abdin et al., 2024) as verifiers and Llama-68m (Miao et al., 2024) as the drafter trained with various KD. Table 10 summarizes that the inference speedup of drafter with DISTILLM-2 surpasses other drafter models, including trained with SFT and DistiLLM for both Phi3.5-mini and Phi3-medium verifiers. These results indicate that the DISTILLM-2 enables higher token-level alignment of distribution compared to other LLM distillation baselines, including Zhou et al. (2024b).

## 7. Conclusion

In this work, we introduce DISTILLM-2, a novel distillation framework for large language models that combines contrastive loss, curated data, and curriculum-based learning. By differentiating teacher and student outputs, our method overcomes key limitations of traditional distillation and achieves stronger alignment and generalization. Extensive experiments across instruction following, mathematical reasoning, and code generation show that DISTILLM-2 delivers state-of-the-art performance with improved sample efficiency, reducing reliance on expensive preference-labeled data. Leveraging high-quality explanations and contrastive objectives further enhances reasoning ability and robustness in both language and vision-language models. We believe DISTILLM-2 lays the groundwork for future advances in efficient model alignment and multi-modal learning, enabling more accessible and capable AI systems.

## Acknowledgements

JK, SK, and SY were (partially) supported by Institute of Information & communications Technology Planning & Evaluation (IITP) grant funded by the Korea government (MSIT) (No. RS-2019-II190075, Artificial Intelligence Graduate School Program (KAIST)) and the Institute of Information & communications Technology Planning & Evaluation (IITP) grant funded by the Korea government (MSIT) (No. RS-2024-00457882, AI Research Hub Project) and Institute of Information & communications Technology Planning & Evaluation (IITP) grant funded by the Korea government (MSIT) (No. 2022-0-00871, Development of AI Autonomy and Knowledge Enhancement for AI Agent Collaboration).

## Impact Statement

This paper presents work whose goal is to advance the field of machine learning, specifically efficiency and efficacy of LLMs-based system. We also have shown four possible applications (**but not limited to**) as follows:

- **High-performed reference model for RLHF** (§6.1) **:** Our DISTILLM-2 can replace vanilla SFT ahead of PPO with a reward model or DPO (or IPO, SimPO; Meng et al. 2024) with chosen and rejected response pairs. While our experiments in Table 6 are conducted on DPO, we believe that this approach can be expanded to various types of preference alignment methods, which can significantly contribute to building safe AI.

- **Extension to multi-modal LLMs** (§6.2) **:** While our DISTILLM-2 is primarily evaluated on LLMs, we also demonstrate its potential adaptability to VLMs in Table 7. Since many multimodal large language models (MLLMs; Dai et al. 2023; Tang et al. 2024) are built on LLMs, we believe that DISTILLM-2 can be effectively applied to a wide range of MLLMs.

- **Recovering the compressed LLMs** (§6.3) **:** While network pruning (Ko et al., 2023b) and quantization (Shao et al., 2024) have significantly improved the efficiency of LLMs, they often come at the cost of inherent performance degradation compared to the uncompressed original models. Our experiments demonstrate that DISTILLM-2 can substantially enhance the performance of quantized LLMs, making them highly competitive with their original counterparts. We believe that DISTILLM-2 can further improve the effectiveness of compressed models, such as quantized LLMs, while also being applicable to other compression techniques.

- **Enhancing Inference speed of LLMs** (§6.4) **:** We also show that DISTILLM-2 can significantly enhance the efficacy of speculative decoding (Chen et al., 2023; Leviathan et al., 2023) by improving the alignment between draft and target models. We believe that DISTILLM-2 can be beneficial for systems that utilize diverse models within a single framework to improve both efficiency and efficacy.

While we have already demonstrated the applicability of our work across various domains, we believe that it can be utilized in an even broader range of fields. For example, the reasoning ability showcased by DeepSeek-R1 (Guo et al., 2025) could be integrated with our methodology to yield more impactful results for sLMs. We encourage future research to explore and discuss the broader implications of this work across diverse domains.

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

# DISTILLM-2: A Contrastive Approach Boosts the Distillation of LLMs

## Supplementary Material

## A. Additional Related Works

**KD in LLMs.** Recently, several works have pioneered the KD for LLMs (Gu et al., 2024; Agarwal et al., 2024; Ko et al., 2024). Unlike small BERT-based models, which have focused on intermediate layer distillation (Wang et al., 2020; Ko et al., 2023a), most works on LLMs have focused on logit-based distillation due to their large number of parameters. Gu et al. (2024) proposed a policy gradient-based method addressing the high variance issues in RL-based methods. Agarwal et al. (2024) propose on-policy approach of SGO with diverse objectives like RKLD and JSD. Based on these pioneer, numerous works (Xu et al., 2024b; Zhang et al., 2024; Li et al., 2025) continuously studied to improve the performance of KD in LLMs. Li et al. (2025) filter out long-tail noise by utilizing top-k teacher and student logits and leverage internal logit ranking information by constructing logit differences. Zhang et al. (2024) introduced the dual-space knowledge distillation (DSKD) framework, which unifies the output spaces of the two models for KD. Similar to our work, Wu et al. (2025) provided adaptive KL to balance their early-stage behaviors of KL and RKL, however, they do not consider about the data perspective of LLM distillation.

**Contrastive approach.** Actor-critic RLHF frameworks (Christiano et al., 2017; Stiennon et al., 2020; Bai et al., 2022; Ouyang et al., 2022) seeks to align language models to human preferences, but is often unstable during training and memory-intensive (requiring the policy model and reward model to be on device simultaneously). To mitigate this, several algorithms (Rafailov et al., 2023; Azar et al., 2024; Ko et al., 2025), such as direct preference optimization (DPO; Rafailov et al. 2023) and sequence likelihood calibration (SLiC-HF; Zhao et al. 2023), learn the contrastive preference in the offline setting using a closed-form loss function without the need for an critic/reward model. (Azar et al., 2024) argued that without regularization, a policy can easily overfit to deterministic preferences and introduced identity preference optimization (IPO) to directly optimize offline preference probabilities with regularization.

## B. Derivation for Mathematical Analysis

### B.1. Mathematical Explanation on Behavior of KL and RKL

Here, we provide mathematical explanation for (S)KL and (S)RKL showed in Figure 1(a) and (b). Formally, the $f$-divergence of two distributions is defined as

$$D_f(p^1, p^2) = \mathbb{E}_{\mathbf{y} \sim p^1} \left[ f \left( \frac{p^1(\mathbf{y}|\mathbf{x})}{p^2(\mathbf{y}|\mathbf{x})} \right) \right] := \mathbb{E}_{p^1} \left[ f \left( \frac{dp^1}{dp^2} \right) \right],$$

where $dp^1$ and $dp^2$ are the probability densities of probability $p^1$ and $p^2$. The KL is a $f$-divergence generated by $f(t) = t \log t$ and RKL is a $f$-divergence by $f(t) = -\log t$. From Ko et al. (2024), the $\alpha$-skew KL divergence is a $f$-divergence generated by $f^{(\alpha)}(t) = t \log \left( \frac{t}{\alpha t + 1 - \alpha} \right)$ and $\alpha$-skl RKL is is a $f$-divergence generated by $f^{(\alpha)}(t) = -\log \left( (1-\alpha)t + \alpha \right)$. Based on these property, we provide detailed explanation for the empirical observation.

**Pulling-up effect of (S)KL:** By taking $f(t) = t \log t$, we have $\lim_{t \to \infty} f(t) = +\infty$. Because $p(\cdot|\boldsymbol{x}) \in (0, 1)$ and $q_\theta(\cdot|\boldsymbol{x}) \in (0, 1)$, $q_\theta(\cdot|\boldsymbol{x})$ cannot be too small when $p(\cdot|\boldsymbol{x})$ is significantly greater than 0. As a result, $q_\theta(\cdot|\boldsymbol{x})$ is encouraged to "pull up" its values where $p(\cdot|\boldsymbol{x})$ is large. Similarly, for SKL, by taking $f(t) = t \log \left( \frac{t}{\alpha t + 1 - \alpha} \right)$, we also have $\lim_{t \to \infty} f(t) = +\infty$. Thus, SKL also benefits from the same "pulling-up" property of the KL-like term.

**Pushing-down effect of (S)RKL:** By taking $f(t) = -\log t$, we have $\lim_{t \to 0^+} f(t) = +\infty$, which means $q_\theta(\cdot|\boldsymbol{x})$ should be small when $p(\cdot|\boldsymbol{x})$ is small. As a result, $q_\theta(\cdot|\boldsymbol{x})$ is encouraged to "push down" its values where $p(\cdot|\boldsymbol{x})$ is near to zero. Similarly, for SRKL, by taking $f(t) = -\log \left( (1-\alpha)t + \alpha \right)$, we also have $\lim_{t \to 0^+} f(t) = +\infty$. Thus, SRKL also benefits from the same "pushing-down" property of RKL-like term.

While Wu et al. (2025) provided a similar observation, their explanation only holds for a unimodal Gaussian distribution, whereas ours applies to a more general problem setup. Additionally, our explanation provides mathematical intuition for both S(R)KL and (R)KL.

## B.2. Derivation for Remark 1

Based on the definitions of SKL and SRKL, we have

$$D_{\text{SKL}}^{(\alpha)}(\boldsymbol{x}, \boldsymbol{y}_t) + D_{\text{SKL}}^{(\alpha)}(\boldsymbol{x}, \boldsymbol{y}_t) = p(\boldsymbol{y}_t|\boldsymbol{x}) \log \frac{p(\boldsymbol{y}_t|\boldsymbol{x})}{\tilde{q}_\theta(\boldsymbol{y}_t|\boldsymbol{x})} + q_\theta(\boldsymbol{y}_s|\boldsymbol{x}) \log \frac{q_\theta(\boldsymbol{y}_s|\boldsymbol{x})}{\tilde{p}_\theta(\boldsymbol{y}_s|\boldsymbol{x})}, \tag{8}$$

$$= \mathbb{E}_{p(\boldsymbol{y}_t|\boldsymbol{x})} \left[ \log \frac{p(\boldsymbol{y}_t|\boldsymbol{x})}{\tilde{q}_\theta(\boldsymbol{y}_t|\boldsymbol{x})} \right] + \mathbb{E}_{q_\theta(\boldsymbol{y}_s|\boldsymbol{x})} \left[ \log \frac{q_\theta(\boldsymbol{y}_s|\boldsymbol{x})}{\tilde{p}_\theta(\boldsymbol{y}_s|\boldsymbol{x})} \right]. \tag{9}$$

Furthermore, as $\boldsymbol{y}_t$ and $\boldsymbol{y}_s$ are independent, the following holds by the linearity of expectation:

$$\mathbb{E}_{p(\boldsymbol{y}_t|\boldsymbol{x})} \left[ \log \frac{p(\boldsymbol{y}_t|\boldsymbol{x})}{\tilde{q}_\theta(\boldsymbol{y}_t|\boldsymbol{x})} \right] + \mathbb{E}_{q_\theta(\boldsymbol{y}_s|\boldsymbol{x})} \left[ \log \frac{q_\theta(\boldsymbol{y}_s|\boldsymbol{x})}{\tilde{p}_\theta(\boldsymbol{y}_s|\boldsymbol{x})} \right] = \mathbb{E}_{\boldsymbol{y}_t \sim p(\boldsymbol{y}_t|\boldsymbol{x}), \boldsymbol{y}_s \sim q_\theta(\boldsymbol{y}_s|\boldsymbol{x})} \left[ \log \frac{p(\boldsymbol{y}_t|\boldsymbol{x})}{\tilde{q}_\theta(\boldsymbol{y}_t|\boldsymbol{x})} + \log \frac{q_\theta(\boldsymbol{y}_s|\boldsymbol{x})}{\tilde{p}_\theta(\boldsymbol{y}_s|\boldsymbol{x})} \right] \tag{10}$$

$$= \mathbb{E}_{\boldsymbol{y}_t \sim p(\boldsymbol{y}_t|\boldsymbol{x}), \boldsymbol{y}_s \sim q_\theta(\boldsymbol{y}_s|\boldsymbol{x})} \left[ \log \frac{p(\boldsymbol{y}_t|\boldsymbol{x})}{\tilde{q}_\theta(\boldsymbol{y}_t|\boldsymbol{x})} - \log \frac{\tilde{p}_\theta(\boldsymbol{y}_s|\boldsymbol{x})}{q_\theta(\boldsymbol{y}_s|\boldsymbol{x})} \right]. \tag{11}$$

From this, we can verify that our Equation 5 can be interpreted as Equation 6, which behaves similarly to the contrastive approach defined in DPO (Rafailov et al., 2023).

## B.3. First-order Approximation for Mercator Series

From the Mercator series expansion, following hold:

$$\log(1 + x) = \sum_{n=1}^{\infty} (-1)^{n+1} \cdot \frac{x^n}{n} = x - \frac{x^2}{2} + \frac{x^3}{3} - \cdots, \tag{12}$$

where the series converges to the natural logarithm whenever $-1 < x \le 1$.

By substituting $p(\boldsymbol{y}|\boldsymbol{x}) - 1$ into x, we can write as follows,

$$\log p(\boldsymbol{y}|\boldsymbol{x}) = (p(\boldsymbol{y}|\boldsymbol{x}) - 1) - \frac{(p(\boldsymbol{y}|\boldsymbol{x}) - 1)^2}{2} + \frac{(p(\boldsymbol{y}|\boldsymbol{x}) - 1)^3}{3} - \cdots. \tag{13}$$

Since the softmax outputs of LLMs, $p(\boldsymbol{y}|\boldsymbol{x})$, satisfy $0 < p(\boldsymbol{y}|\boldsymbol{x}) \le 1$ by the definition of probability, it follows that $-1 < p(\boldsymbol{y}|\boldsymbol{x}) - 1 \le 0$. This holds because the softmax function outputs strictly positive values due to the exponential transformation of real-valued inputs.

Hence, from the first-order Mercator series expansion approximation, we have

$$\log \frac{p(\boldsymbol{y}|\boldsymbol{x})}{\alpha p(\boldsymbol{y}|\boldsymbol{x}) + (1 - \alpha) q_\theta(\boldsymbol{y}|\boldsymbol{x})} = \log p(\boldsymbol{y}|\boldsymbol{x}) - \log (\alpha p(\boldsymbol{y}|\boldsymbol{x}) + (1 - \alpha) q_\theta(\boldsymbol{y}|\boldsymbol{x})) \tag{14}$$

$$= [(p(\boldsymbol{y}|\boldsymbol{x}) - 1) - (\alpha p(\boldsymbol{y}|\boldsymbol{x}) + (1 - \alpha) q_\theta(\boldsymbol{y}|\boldsymbol{x}) - 1)] - \cdots \tag{15}$$

$$\simeq (1 - \alpha) p(\boldsymbol{y}|\boldsymbol{x}) + (1 - \alpha) q_\theta(\boldsymbol{y}|\boldsymbol{x}) = (1 - \alpha) \cdot (p(\boldsymbol{y}|\boldsymbol{x}) - q_\theta(\boldsymbol{y}|\boldsymbol{x})), \tag{16}$$

which holds for $0 \le \alpha \le 1$. By choosing first-order approximation, we can express the S(R)KL as a closed-form function of $\alpha$, $p(\boldsymbol{y}|\boldsymbol{x})$, and $q_\theta(\boldsymbol{y}|\boldsymbol{x})$ which enables to compute proper $\alpha$ for each sample easily. Instead, as we compromise approximation error for either $p(\boldsymbol{y}|\boldsymbol{x}) \ll 1$ or $q_\theta(\boldsymbol{y}|\boldsymbol{x}) \ll 1$, we apply mini-batch wise allocation and clipping for improving the stability of implementation of curriculum-based approach. For clipping, we utilize upper and lower bound as 0.1 and 0.01.

## C. Detailed Experimental Setup

We elaborate the detailed experimental setup regarding the datasets used (§C.1), training details (§C.2), and evaluation details (§C.3). For all experiments, we implement DISTILLM-2 using the `trl` framework, as well as for other baselines, including GKD (Agarwal et al., 2024) and SKD (Xu et al., 2024b).

**C.1. Dataset Description**

We apply DISTILLM-2 on instruction-following, math reasoning, and code generation datasets. We provide detailed descriptions of the datasets used.

- **UltraChat200k** (instruction-following; Tunstall et al. 2023 [4]): This is a heavily filtered version of UltraChat (Ding et al., 2023), originally used to train Zephyr-7B-$\beta$ (Tunstall et al., 2023). It is obtained from the original version, which consists of 1.4M dialogues generated by ChatGPT and spans a wide range of topics, by removing the dialogues that contain grammatical errors or where the assistant replies with phrases like "I do not have emotions" or "I don't have opinions."

- **AlpacaEval** (instruction-following; Dubois et al. 2024 [5]): This dataset is slight modifications (or simplification) of the AlpacaFarm evaluation set. Dubois et al. (2024) first merged the instruction and input fields into a single instruction field. This affects 1/4 of the examples in the AlpacaFarm evaluation set, all of which are from the Self-Instruct (Wang et al., 2023). This dataset contains 805 challenging questions.

- **Evol-Instruct Evaluation** (instruction-following; Xu et al. 2024a [6]): Evol-Instruct (Xu et al., 2024a) contains 218 questions, spanning multiple topics generated using the Evol-Instruct procedure.

- **UltraFeedback** (instruction-following; Cui et al. 2024; Tunstall et al. 2023 [7] [8]): This is a large-scale, fine-grained, and diverse preference dataset used for training powerful reward models and critic models. Cui et al. (2024) collected about 64k prompts from diverse resources, including UltraChat, ShareGPT, and Evol-Instruction (Xu et al., 2024a). They used these prompts to query multiple LLMs, generating four different responses for each prompt. The responses were annotated using GPT-4 to collect high-quality preferences based on instruction-following, truthfulness, honesty, and helpfulness.

- **MetaMathQA** (mathematical reasoning; Yu et al. 2024a [9]): MetaMathQA is a dataset introduced in Yu et al. (2024a) to improve mathematical reasoning in large language models. It is created through question bootstrapping, where mathematical problems are rewritten from multiple perspectives, including forward reasoning, backward reasoning, and rephrasing.

- **GSM8K** (mathematical reasoning; Cobbe et al. 2021 [10]): GSM8K (Grade School Math 8K) is a dataset comprising 8.5K high-quality, linguistically diverse grade school math word problems. It is designed to facilitate question answering on fundamental mathematical problems that involve multi-step reasoning.

- **MATH** (mathematical reasoning; Hendrycks et al. 2021 [11]): This dataset code generates mathematical question-and-answer pairs covering various question types at approximately school-level difficulty. It is designed to evaluate learning models' mathematical comprehension and algebraic reasoning abilities.

- **WizardCoder** (code generation; Luo et al. 2024 [12]): WizardCoder dataset is constructed using the Evol-Instruct method, which refines and expands existing code instruction datasets. The process starts with Code Alpaca, a 20K-sample instruction-following dataset, and iteratively applies instruction evolution techniques to generate progressively more complex training data. These modifications include adding constraints, increasing reasoning steps, providing misleading code, and enforcing time-space complexity requirements. The final dataset consists of approximately 78K evolved samples, which are used to fine-tune the StarCoder model, significantly improving its performance on code generation benchmarks.

- **HumanEval** (code generation; Chen et al. 2021 [13]): The HumanEval dataset, released by OpenAI, consists of 164 programming problems, each containing a function signature, docstring, body, and multiple unit tests. These problems were manually crafted to ensure they were not part of the training data for code generation models.

- **MBPP** (code generation; Austin et al. 2021 [14]): The benchmark includes approximately 1,000 crowd-sourced Python programming problems, designed for entry-level programmers and covering programming fundamentals, standard library

---

[4] https://huggingface.co/datasets/HuggingFaceH4/ultrachat_200k
[5] https://huggingface.co/datasets/tatsu-lab/alpaca_eval
[6] https://github.com/nlpxucan/WizardLM/blob/main/WizardLM/data/WizardLM_testset.jsonl
[7] https://huggingface.co/datasets/openbmb/UltraFeedback
[8] https://huggingface.co/datasets/HuggingFaceH4/ultrafeedback_binarized
[9] https://huggingface.co/datasets/meta-math/MetaMathQA
[10] https://huggingface.co/datasets/openai/gsm8k
[11] https://huggingface.co/datasets/deepmind/math_dataset
[12] https://huggingface.co/datasets/nickrosh/Evol-Instruct-Code-80k-v1
[13] https://huggingface.co/datasets/openai/openai_humaneval
[14] https://huggingface.co/datasets/google-research-datasets/mbpp

functions, and more. Each problem features a task description, a code solution, and three automated test cases. As noted in the paper, a portion of the dataset has been manually verified.

- **RLAIF-V-Dataset** (visual question answering; Yu et al. 2024b [15]): RLAIF-V-Dataset is a comprehensive multimodal feedback dataset featuring 83,132 preference pairs with high-quality annotations. The instructions are sourced from a diverse selection of datasets, including MSCOCO, ShareGPT-4V, MovieNet, Google Landmark v2, VQA v2, OKVQA, and TextVQA. Additionally, authors incorporate image description prompts from RLHF-V, utilizing them as long-form image-captioning instructions.

- **OK-VQA** (visual question answering; Marino et al. 2019 [16]): OK-VQA is a large-scale visual question answering (VQA) dataset with over 14,000 questions that require external knowledge to answer. Unlike traditional VQA datasets, it challenges models to retrieve and integrate external information rather than relying solely on image content. As a diverse and challenging dataset, OK-VQA surpasses previous knowledge-based VQA benchmarks in scale, making it a crucial resource for advancing AI reasoning capabilities.

- **TextVQA** (visual question answering; Singh et al. 2019 [17]): TextVQA is a dataset designed to benchmark visual reasoning based on text in images. To answer TextVQA questions, models must read and interpret text within images, integrating this textual information into their reasoning process. Unlike traditional VQA tasks, TextVQA requires models to handle both visual and textual modalities, making it a unique challenge in multi-modal learning.

### C.2. Training Details

Here, we describe the hyperparameters and implementation details for training with DISTILLM-2. Our hyperparameters are shown in Table 11. For all experiments, we utilize LoRA (low-rank adaptation; Hu et al. 2022), which one of the most popular parameter-efficient fine-tuning techniques, for training efficiency. For all models, we use the maximum batch size that fits on **4 NVIDIA A100 80GB GPUs**, while matching the effective batch size with 128 by considering the batch size and gradient accumulation. For all experiments in §4, we first train the student models on training datasets with ground-truth responses using SFT, and then conduct KD for LLMs. Instead, we also provide the results for the student models initialized from instruction models with Gemma-2-2B-it (DeepMind et al., 2024) in Appendix D.2. Unlike the previous version (Ko et al., 2024), we do not use language modeling loss on pre-training corpus for all experiments.

*Table 11.* Hyperparameter values used in DISTILLM-2 experiments in §4.

| Hyperparameter | Instruction-following | Mathematical Reasoning | Code generation |
|---|---|---|---|
| Fine-tuning method | | LoRA ($r = 16$) | |
| Target module for LoRA | | all linear layers for self-attention and MLP layers in Transformer network | |
| Learning rate | | $5.0 \times 10^{-5}$ | |
| Effective Batch Size | | 128 | |
| # Epochs | 3 epochs | 2 epochs | 2 epochs |
| Initial $\alpha_0$ | | 0.1, we do not use curriculum-based update in 1st epoch. | |
| Clipping value $\beta_0$ | | 0.5 | |

### C.3. Evaluation

**Instruction-following.** For evaluating the trained LLMs, we applied a single NVIDIA A100 80GB GPU for sampling the responses from each model using a temperature of 0.8, top-p value of 0.95, a max-length limit of 512. For LLM-as-a-Judge (Zheng et al., 2023) evaluation, we use a pairwise comparison prompt which depicted in Figure 4 with setting the temperature of 0.7. For AlpacaEval, we conducted pairwise comparisons against responses from `text-davinci-003`, which have been officially released. For Evol-Instruct and UltraFeedback, we compared generated responses to those from `gpt-3.5-turbo`, which were produced internally. To avoid position bias, we average the results by switching the order of the compared responses.

**Mathematical Reasoning & Code generation.** For evaluating the trained student models, we applied a single NVIDIA A100 80GB GPU for sampling the responses from the each model using greedy sampling, a max-length limit of 1024. Specifically, for code generation, our evaluation is conducted on EvalPlus framework (Liu et al., 2023).

[15] https://huggingface.co/datasets/openbmb/RLAIF-V-Dataset
[16] https://huggingface.co/datasets/HuggingFaceM4/OK-VQA
[17] https://huggingface.co/datasets/facebook/textvqa

---

[System]

Please act as an impartial judge and evaluate the quality of the responses provided by two AI assistants to the user question displayed below. You should choose the assistant that follows the user's instructions and answers the user's question better. Your evaluation should consider factors such as the helpfulness, relevance, accuracy, depth, creativity, and level of detail of their responses. Begin your evaluation by comparing the two responses and provide a short explanation. Avoid any position biases and ensure that the order in which the responses were presented does not influence your decision. Do not allow the length of the responses to influence your evaluation. Do not favor certain names of the assistants. Be as objective as possible. After providing your explanation, output your final verdict by strictly following this format: "[[A]]" if assistant A is better, "[[B]]" if assistant B is better, and "[[C]]" for a tie.

[User Question]
{question}

[The Start of Assistant A's Answer]
{answer_a}
[The End of Assistant A's Answer]

[The Start of Assistant B's Answer]
{answer_b}
[The End of Assistant B's Answer]

---

*Figure 4.* The pairwise comparison prompt introduced in LLM-as-a-Judge (Zheng et al., 2023).

## D. Additional Experimental Results

### D.1. Comparison on On-policy Setup

**Setup.** We also compare our methods with on-policy manner algorithms, in the code-base of DistiLLM. We compare the recent on-policy distillation baselines, including ImitKD (Lin et al., 2020), MiniLLM (Gu et al., 2024), GKD. We also provide the results for the adaptive on-policy setup for DistiLLM and DISTILLM-2. Also, we conducted experiments with DPKD and DPO. We follow the experimental setup of Ko et al. (2024), which trained GPT-2 (Radford et al., 2019) on databricks-dolly-15k (Conover et al., 2023). All hyper-parameter setups are from (Ko et al., 2024). Note that we apply same experimental setup for the result in Figure 1(c).

*Table 12.* Application of "batched" on-policy setup compared to fully on-policy and off-policy setup. We evaluate the student models in databricks-dolly-15k test set with ROUGE-L following Ko et al. (2024).

| Size | MiniLLM | GKD | DPO | DPKD | DistiLLM | DISTILLM-2 |
|---|---|---|---|---|---|---|
| on-policy | 23.84 | 23.75 | - | 6.85 | 26.12 | **26.37** |
| *batched* on-policy (ours) | 23.75 | 23.21 | 23.42 | 6.43 | 26.11 | **26.20** |
| off-policy | 23.41 | 22.89 | 22.78 | 6.43 | 26.12 | **26.13** |

**Results.** Table 12 show that our "batched" on-policy setup, which shares the same takeaway as the adaptive off-policy approach in Ko et al. (2024), does not suffer from severe performance degradation despite its significant efficiency. Also, we observe that DPKD (Li et al., 2024b) performs much worse than its reported values, as it is prone to reward hacking, as we reported in §3.1. Hence, we decide not to include DPKD in our main baselines in §4. While we provide results for DPO, except in the on-policy setup, its performance is worse than DISTILLM-2 but better than DPKD.

### D.2. LLM distillation with Inst models

While the results in Table 2 focus on the base model as the student model, we also provide results using the inst model as the student model, specifically Gemma-2-2B-it (DeepMind et al., 2024) with Gemma-2-9B-SimPO (Meng et al., 2024) as the teacher model. We use the same experimental setup as the base model, except that we train the inst model for only a single epoch of 200 training iterations.

*Table 13.* Comparison of the teacher model ($\mathcal{M}_T$) and student models with different KD methods. Note that DISTILLM-2 achieve higher performance than teacher in UltraFeedback evaluation.

| | $\mathcal{M}_T$ | $\mathcal{M}_S$ | GKD | DistiLLM | DISTILLM-2 |
|---|---|---|---|---|---|
| Evol-Inst | 88.76 | 76.80 | 79.57 | 80.28 | **85.10** |
| UltraFeedback | 85.90 | 79.52 | 81.94 | 85.56 | **88.26** |

Overall, the student models in Table 13 achieve higher performance compared to the base models (+SFT) in Table 2. Notably, our DISTILLM-2 achieves even higher performance in the UltraFeedback evaluation compared to other student models, demonstrating that LLM distillation remains effective for recent state-of-the-art models. We belive that these results stem from the fast convergence of the contrastive approach in our DISTILLM-2, even with very limited training iterations.

