# OpenReview forum: "DistiLLM-2: A Contrastive Approach Boosts the Distillation of LLMs"
_ICML.cc/2025/Conference — ICML 2025 oral_

### Official Review · Reviewer_WSDW · 2025-03-07

**Overall Recommendation:** 4

**Summary:**

The paper introduces DistiLLM-2, a novel approach to distilling knowledge from large language models (LLMs) into smaller, more efficient student models.  DistiLLM-2 employs a contrastive approach that leverages the synergy between loss formulations and data types, simultaneously increasing the likelihood of teacher responses and decreasing that of student responses.  The authors claim that this method achieves state-of-the-art performance for student LLMs across various tasks, including instruction-following, mathematical reasoning, and code generation.  Additionally, DistiLLM-2 supports diverse applications such as preference alignment and vision-language extensions.  The key innovation lies in the contrastive loss function, which applies distinct loss functions to different types of training samples, effectively incorporating the synergy between loss formulations and data perspectives.  The paper also introduces optimized dataset curation strategies and curriculum-based adaptive loss mechanisms to further enhance the distillation process.

**Claims And Evidence:**

Supported Claims:
- Improved performance across tasks
- Contrastive approach effectiveness is analytically supported
- Data curation strategy for teacher and student generations
- Showed impact of datset size
- Showed generalization to new tasks including vision

**Essential References Not Discussed:**

none that i can think of

**Experimental Designs Or Analyses:**

As mentioned in the methods, it looks good.

**Methods And Evaluation Criteria:**

Yes.Methods:

Contrastive Loss Function: The core idea of using a contrastive loss function that leverages the synergy between loss formulations and data types is well-motivated and addresses what the authors see as the limitations of previous distillation methods. Paper also introduced: Data Curation Strategy and Curriculum-based Adaptive Learning

Evaluation Criteria:

Benchmark Evals: AlpacaEval, Evol-Instruct, UltraFeedback, GSM8K, MATH, HumanEval, and MBPP are widely recognized and represent relevant tasks for evaluating LLM capabilities.
LLM-as-a-Judge: The use of LLM-as-a-Judge for evaluating instruction-following tasks provides a robust and comprehensive assessment of the generated responses.
Pass@k for Code Generation: The use of Pass@k as an evaluation metric for code generation tasks is standard practice and accurately reflects the best case ability of the student model to generate correct and executable code.
Speculative Decodeing: Is a great proxy for how well a student matches a teacher.

In summary, the proposed methods and evaluation criteria are well-aligned with the problem of LLM distillation.

**Other Comments Or Suggestions:**

-

**Other Strengths And Weaknesses:**

-

**Questions For Authors:**

-

**Relation To Broader Scientific Literature:**

distillation losses are extremely important to a large set of problems including RL, specdec, teaching student models.

**Theoretical Claims:**

AppendixB had all the proofs; all very mathy... I'm not 100% sure I'm super qualified to look thru this, but here we go...

- Behavior of KL and RKL: The paper provides a mathematical explanation for the behavior of KL and RKL, including the "pulling-up" effect of KL and the "pushing-down" effect of RKL.  The explanation made sense / didn't seen to have issues.

- Derivation for Remark 1: The paper presents a derivation for Remark 1, which establishes a mathematical connection between the proposed CALD loss function and DPKD/DPO (Appendix B.2).  I looked thru the derivation; it didn't seen to have issues.

- First-order Approximation for Mercator Series: idk if I'm super qualified to be checking series but it looked fine.

---

> ### Author Rebuttal · Authors · 2025-03-29
>
> We appreciate your constructive comments. We have rephrased the question for ease of reference and provide our corresponding responses below. Look forward to any further discussions.
>
> ***Q1. Inclusion of acceptance rates in speculative decoding analysis for completeness***
>
> **A1.** Thank you for the suggestion. To provide a more concrete answer, we evaluated the acceptance rates using the vLLM [1] framework, which supports detailed speculative decoding metrics. The results of acceptance rates are presented as follows:
>
> |          | SFT   | GKD   | DistiLLM | DistiLLM-2 |
> |----------|-------|-------|----------|------------|
> | Phi-3-medium | 0.412 | 0.464 | 0.469    | 0.487      |
> | Phi-3.5-mini | 0.397 | 0.443 | 0.452    | 0.522      |
>
> We could see that DistiLLM-2 significantly boosted the acceptance rates leading to a higher speed-up via speculative decoding.
>
> [1] Kwon et al., “Efficient Memory Management for Large Language Model Serving with PagedAttention.” SOSP. 2023

---

> > ### Comment · Reviewer_WSDW · 2025-04-08
> >
> > thank you for updating.

---

### Official Review · Reviewer_wyun · 2025-03-08

**Overall Recommendation:** 4

**Summary:**

This paper addresses the critical challenge of compressing large language models (LLMs) for practical deployment by focusing on knowledge distillation (KD). The authors highlight the limitations of existing KD approaches, which primarily focus on either optimizing loss functions (like Kullback-Leibler divergence or skew KL) or curating training data (teacher-generated vs. student-generated outputs).
They also point out that current methods often overlook the synergistic relationship between loss formulations and data types, limiting the potential performance gains in student models. Additionally, the authors note the rise of contrastive learning methods (like DPO) but observe that direct application of DPO to KD suffers from reward hacking issues.

To address these shortcomings, the paper introduces DISTILLM-2, a novel contrastive approach for KD of LLMs that builds upon DistilLLM. Its main contributions are as follows:

(1) The introduction of a contrastive approach with asymmetric loss dynamics (CALD): This involves analyzing and leveraging the behavior of forward and reverse KL (and SKL) during training, and applying distinct loss functions to different training samples.

(2) Optimized dataset curation and curriculum-based adaptive loss mechanisms: These enhancements provide practical guidelines for their contrastive approach.

(3) Demonstrated performance: The paper claims state-of-the-art performance for small language models (sLMs) across various tasks, including instruction-following, reasoning, and code generation, and also demonstrates the methods versatility by applying it to preference alignment and vision-language models.

**Claims And Evidence:**

I feel the claims made in the submission are supported by clear and convincing evidence.

**Essential References Not Discussed:**

No

**Experimental Designs Or Analyses:**

I went through the experimental design and analysis of the paper and it looks sound and valid to me.

**Methods And Evaluation Criteria:**

The proposed methods and evaluation criteria do make sense for the problem at hand.

**Other Comments Or Suggestions:**

This is minor, but I was a bit confused when reading Lines 152-156 — maybe the authors can rephrase their statement there.

**Other Strengths And Weaknesses:**

Strength: The authors have conducted extensive experimentation both to demonstrate the effectiveness of their techniques but also to motivate their ideas.
Weakness: The paper does not have any major weaknesses.

**Questions For Authors:**

I have no questions for the authors.

**Relation To Broader Scientific Literature:**

This work builds upon DistiLLM (Ko et. al., 2024) which is a state-of-the-art approach for distilling LLMs.  The paper proposes a new approach for distilling LLMs, by both combining and refining several known techniques, and by introducing new ideas. It seems to be achieving significant improvements over the state-of-the-art across various benchmarks.

**Theoretical Claims:**

The paper does not have any major theoretical contributions — only a few theoretical remarks / observations. (I have not checked the calculations behind these remarks, but they seem straightforward and they're not a major part of this paper anyway.)

---

> ### Author Rebuttal · Authors · 2025-03-29
>
> We thank the reviewer for their constructive feedback. Below we address each concern in detail. Look forward to any further discussions.
>
> ***Q1. Lines 152–156 might be confusing – rephrasing suggested***
>
> A1. Thank you for the suggestion. We will revise Lines 152–156 to improve clarity. This paragraph aims to illustrate the relation between contrastive learning in preference alignment and our approach to knowledge distillation: increasing the student model's likelihood of generating outputs aligned with teacher responses, and decreasing it for those resembling weaker student responses, using distinct loss functions.
>
> To make this point clear, we plan to revise the paragraph as follows:
>
> “Similarly, we incorporate this contrastive concept from preference optimization into KD by assigning different loss functions to different types of responses: encouraging the student model to assign higher likelihood to high-quality responses generated by the teacher ($y_t$) -- $q_\theta(y_t|x)$ -- while reducing the likelihood of lower-quality student responses ($y_s$) -- $q_\theta(y_s|x)$ -- that deviates from the teacher.”

---

### Official Review · Reviewer_F2H5 · 2025-03-13

**Overall Recommendation:** 3

**Summary:**

This paper introduces DistiLLM-2, a contrastive approach for LLM distillation, optimizing student models by increasing teacher response likelihood (SKL loss) and decreasing student response likelihood (SRKL loss). It improves data curation, adaptive learning, and curriculum-based loss weighting, outperforming baselines in instruction-following, math reasoning, and code generation.

**Claims And Evidence:**

The claims are supported by clear and convincing evidences.

**Essential References Not Discussed:**

N/A

**Experimental Designs Or Analyses:**

The experiments on general instruction tuning, math, and code domains are sound and valid.

**Methods And Evaluation Criteria:**

The evaluation methods make sense.

**Other Comments Or Suggestions:**

N/A

**Other Strengths And Weaknesses:**

Weakness on clarity:
The presentation of the method can be clearer. For example, what does the term CALD stand for? It comes out in lines 61-62 without any explanation.

**Questions For Authors:**

N/A

**Relation To Broader Scientific Literature:**

This paper provides modifications to the previous on-policy KD methods proposed in MiniLLM[1] and GKD[2] and shows better empirical approaches.


[1] MiniLLM: Knowledge Distillation of Large Language Models.

[2] On-Policy Distillation of Language Models: Learning from Self-Generated Mistakes.

**Theoretical Claims:**

The approximation in Equation 7 seems problematic. In Appendix B.3., the approximation relies on the assumption that $p(y|x) \approx 1$ and $\alpha p(y|x) + (1 − \alpha) q_{\theta}(y|x) \approx 1$. However, when y is a sequence, p(y|x) can be quite small due to the accumulation of per-token probabilities, making the approximation unreasonable. Could the authors provide empirical results on how the values of $p(y|x)$ are distributed on real datasets?

---

> ### Author Rebuttal · Authors · 2025-03-30
>
> We thank the reviewer for their constructive feedback. Below we address each concern in detail. Look forward to any further discussions.
>
> ***Q1. Empirical validation of the approximation in Equation (7) for sequence-level probabilities***
>
> **A1.** Thank you for pointing out the potential mismatch between our first-order Mercator approximation in Equation (7) and actual sequence-level probabilities $p(y|x)$. Indeed, this approximation assumes $p(y|x) \simeq 1$, which can be problematic for long sequences where $p(y|x)$ might vanish.
>
> In our implementation, we compute probabilities at the token level, clip extreme values to avoid outliers, and average them over the sequence, yielding a reasonably scaled alpha that is shared across all tokens.
>
> The corresponding code (in `src/distillm_trainer.py`, lines 1161–1164) is:
> ```
> anchor = (1 - base_alpha_1) * logp_logq
> logps_logqs = (
>     (tea_per_token_logps * loss_mask).sum(-1) / loss_mask.sum(-1)
> ).exp() - (
>     (per_token_logps * loss_mask).sum(-1) / loss_mask.sum(-1)
> ).exp()  # sentence-level
> alpha_1 = torch.clip(
>     1 - anchor / (logps_logqs + 1e-5),
>     min=1e-2, max=base_alpha_1
> ).unsqueeze(-1).unsqueeze(-1)
> ```
> This design makes the approximation **invariant to sequence length**, effectively resolving the concern about very small $p(y|x)$ values in long sequences. Notably, we want to highlight that this first-order approximation has advantage in closed-form $\alpha$ updates for efficient, per-sample adaptation.
>
> Additionally, we provide empirical results of token-level probabilities on teacher ($y_t$) and student ($y_s$) responses to quantify the first-order approximation using the teacher (Mistral-7B) and student (Danube2-1.8B) models. To better characterize the distribution of token-level probabilities, we report the first (Q1), second (Q2, median), and third (Q3) quartiles for both models:
>
> |         | $y_t$  |      |      | $y_s$  |      |      |
> |---------|------|------|------|------|------|------|
> |         | Q1   | Q2   | Q3   | Q1   | Q2   | Q3   |
> | teacher | 0.82 | 0.88 | 0.94 | 0.79 | 0.85 | 0.92 |
> | student | 0.80 | 0.85 | 0.94 | 0.81 | 0.89 | 0.96 |
>
> The results show that most of the token-level probability values are close to 1, which ensures small approximation error of our first-order approximation. We appreciate the reviewer’s insightful comment and will update the paper accordingly to better explain this implementation and its practical implications.
>
> ***Q2. CALD term appears without explanation – unclear terminology***
>
> **A2.** Thanks for the question. CALD stands for **C**ontrastive **A**pproach for **L**LM **D**istillation, which is first introduced in lines 60-61 in our main manuscript. To improve the readability and clarity, we will also define the term at the beginning of Section 3.1.2. We will additionally refine Section 3 and add more descriptive languages right after the definition (Equation (5)).

---

### Official Review · Reviewer_nZhc · 2025-03-15

**Overall Recommendation:** 4

**Summary:**

The paper introduces DISTILLM-2, a novel approach for LLM knowledge distillation. Unlike prior work that applies identical loss functions to both teacher- and student-generated data, DISTILLM-2 leverages a contrastive loss function to explicitly increase the likelihood of teacher responses while decreasing that of student responses. Extensive experiments demonstrate that DISTILLM-2 achieves superior performance across multiple tasks, including instruction-following, mathematical reasoning, and code generation.

**Claims And Evidence:**

The claims in the paper are well-supported by clear and convincing evidence.
Sections 4.1 to 4.3 present experiments across three tasks—general instruction-following, mathematical reasoning, and code generation—using three different LLM teacher-student pairs. The results consistently demonstrate that DISTILLM-2 outperforms baseline KD methods, providing enough empirical validation for its effectiveness.

**Essential References Not Discussed:**

No.

**Experimental Designs Or Analyses:**

Yes, I reviewed the experimental design and analyses, which appear to be sound and well-structured. However, the largest LLM used in the study is only 9B, which raises concerns about the generalizability of the conclusions to larger-scale models. Additional experiments with larger models would strengthen the validity of the findings.

**Methods And Evaluation Criteria:**

Yes

**Other Comments Or Suggestions:**

No

**Other Strengths And Weaknesses:**

Please refer to the raised concern in `Experimental Designs Or Analyses`

**Questions For Authors:**

No

**Relation To Broader Scientific Literature:**

Yes.

**Theoretical Claims:**

Yes

---

> ### Author Rebuttal · Authors · 2025-03-29
>
> Thank you for your insightful feedback. We have rephrased your comments for simpler reference and have included our respective responses. Look forward to any further discussions.
>
> ***Q1. Use of models up to 9B; interest in evaluating scalability to larger models***
>
> Thank you for the insightful comment. In the evaluation, we already demonstrated the effectiveness of DistiLLM-2 over varying sizes of teacher and student models. For teacher models, we have included larger backbones with over 9B parameters, such as **Qwen1.5-14B** in **Appendix D.2** (Table 10) and **Phi-3-Medium-14B** in **Appendix D.4** (Table 12).
>
> For larger student models, we are working on conducting more such experiments. Although we might not be able to report the results in a timely manner during rebuttal due to the resource cost, we will report them in the future.
>
> Mathematically, as explained in Remark 1, DistiLLM-2 proposes a dedicated CALD objective function that could exhibit behavior similar to DPO in preference alignment. Given DPO’s wide validation across diverse architectures and the supporting results from our experiments, we are confident in the generality of the proposed approach.

---

### Decision · Program_Chairs · 2025-05-01

**Decision:**

Accept (oral)

**Comment:**

This paper aims to improve the distillation mechanism for LLM. The paper proposes DistiLLM-2 that leverages a contrastive approach for distillation objectives by employing different loss components for teacher-generated data and student-generated data. In addition, the paper explores adaptive mixing of teacher-student distribution while defining loss function, and proposes a curriculum for adjusting the weight of two loss components in the distillation objective.

All the reviewers agree that the paper makes significant contributions to the SoTA distillation for LLM. The utility of the proposed distillation method is well justified by an extensive empirical evaluation. Furthermore, all the design choices are well supported by a careful analysis. Overall, this paper will be a great addition to ICML.